# POLY-AUTOREGRESSIVE MODELING FOR INTERACTING ENTITIES

## ABSTRACT

We present a simple framework that predicts an agent's future behavior by considering the effects that other interacting agents and entities have on them. We propose to model behavior as a sequence of tokens, each representing the state of an agent at a specific timestep. The core of our approach centers around Poly-Autoregressive models, which predict the future behavior of an agent during interaction by considering the agent's past state history and the state of other agents in the scene. In this paper, we develop the mechanics of Poly-Autoregressive (PAR) modeling and show that this framework applies without any modification to an extensive range of prediction problems that, on the surface, appear as entirely different scenarios, such as human action prediction in social situations, trajectory prediction for autonomous vehicles, and object pose prediction during hand-object interaction.

## 1 INTRODUCTION

The future of large predictive models lies not only in pure language-based tasks confined to the digital space but also in real-world applications that consider multiple agents interacting in the world. To move artificial intelligence from the computer to the real world, we must be able to predict how other agents (human or artificial) are likely to behave. As we know from everyday life, such predictive capabilities are just as handy at a cocktail party as when driving on the road.

In language, predictive models such as LLMs have been quite successful, which is partly enabled by their use of discrete "word" tokens. But what should be the visual video equivalent of a "word" token used for prediction in large language models? Rather than a pixel or a patch, we propose to focus on entities such as a human, an object, or a car as the object of interest, with an associated token "state" that can include data from various modalities, such as location, pose, action, and appearance.

Had our focus been on predicting the outcomes of physical interactions between inanimate objects, such as collisions of a set of billiard balls, we could have taken the approach of constructing a physics simulator from a set of mathematical rules that would perform the prediction of future states for us. Unfortunately, behavior prediction is unlike physics in that we cannot easily simulate it because there is a latent variable about which we know nothing—the internal state of other agents. Instead, we resort to data-driven methods and learn to predict behavior by directly observing large datasets of videos of natural interactions in the wild.

Given these large troves of video and the entity-based state tokens extracted, how should we predict behavior in practice? One popular option is autoregressive prediction (AR), where all the context needed to predict what someone will do in the future are the actions they have taken up to the current moment. Autoregressive prediction takes as input a prefix of tokens as context. At each step, it uses this context and its previous predictions to predict the next token in the sequence (see Figure 1a). However, in social situations for example, the history of a person's past states does not uniquely determine the dynamics of their future states. We must also consider the interactions of said person with other people. For instance, how one drives through a busy city street is not just a result of their intended destination but also the desire to avoid colliding with obstacles and other cars. The vanilla autoregressive model does not capture this dependence on other agents.

To address these concerns, we propose *poly-autoregressive* (PAR) modeling—a simple unifying approach to a surprisingly diverse set of problems that can all be formulated as behavior prediction during interaction. In this paper, we develop the mechanics of the approach and show that this

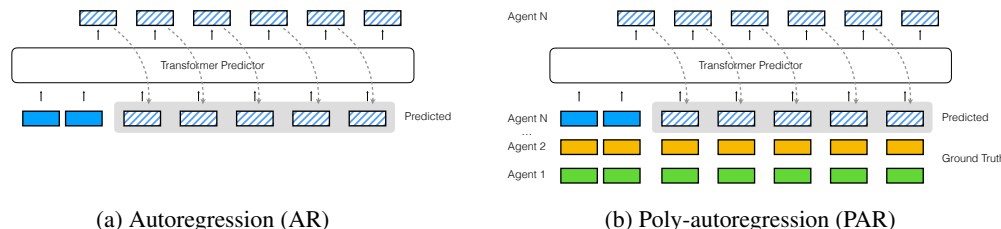

(a) Autoregression (AR)  (b) Poly-autoregression (PAR)

Figure 1: Inference for (a) autoregressive (AR) models and (b) our poly-autoregressive (PAR) model. Solid indicates ground-truth tokens; striped predicted. Colors denotes agent identity. Compared to AR models, PAR model, predicts a new token at every time step, but takes other agent's tokens as inputs.

single general formulation can explain several seemingly different prediction tasks. Our framework considers the influence of interaction with others on one's behavior. We model behavior as a temporal sequence of states and predict an agent's future behavior conditioned on their and their interacting partners' past behavior. By considering other agents' behavior, we demonstrate that our approach significantly improves upon the ambiguous problem of single-agent prediction in interactive settings.

We design our poly-autoregressive prediction framework as a transformer prediction model. Transformers have shown great success in language modeling and naturally lend themselves to behavior prediction. In an interaction scenario of N agents, the transformer predictor predicts the future behavior of the Nth agent conditioned on their past behavior and the behavior of the other N-1 agents (See Figure 1b). We model behavior in a scenario-specific fashion, considering different data modalities (such as action, acceleration, and pose) for each agent at each time step.

We focus our analysis on three seemingly different interactive problems, all of which we can model via the same simple poly-autoregressive prediction framework and implement using the same 4M parameter transformer *without any modifications to the base framework or architecture*: action prediction in social settings, trajectory prediction for autonomous vehicles in busy roads, and object pose prediction during hand-object interaction. In all settings, we demonstrate that taking the other agents in the scene into account results in significantly better performance than predicting the future behavior of one agent in isolation.

## 2 RELATED WORK

**Autoregressive models.** Autoregressive modeling has a rich history in information theory and deep learning, tracing back to Shannon's 1951 paper on language prediction (Shannon, 1951) and Attneave's 1954 study on visual perception (Attneave, 1954). These foundational works laid the groundwork for modern applications in deep learning. (Larochelle & Murray, 2011) revisited interest in neural autoregressive models, and for continuous-valued modeling by (Gregor et al., 2014) and (Theis & Bethge, 2015). (Van Den Oord et al., 2016) developed PixelRNN and PixelCNN, which generates one pixel at a time, using RNNs and CNN respectively.

With the development in transformer models (Vaswani, 2017), image transformer (Parmar et al., 2018) and vision transformer (Dosovitskiy, 2020) for pixels and the GPT family of models (Radford, 2018; Radford et al., 2019; Brown, 2020) natural language processing were developed, which demonstrated the power of large-scale unsupervised autoregressive pre-training. Recent research has focused on multimodal learning, exemplified by the Flamingo (Alayrac et al., 2022) or LlaVa (Liu et al., 2023) models, which combine vision and language processing capabilities, illustrating the versatility of autoregressive models across various domains in artificial intelligence. While these approaches operate on image patches, we operate on symbolic representations extracted from video. A recent approach to humanoid locomotion (Radosavovic et al., 2024) frames the problem as autoregressive next-token prediction that operates on two types of continuous tokens: observations and actions. This approach projects continuous tokens to the hidden dimension and uses a shifted loss similar to the next-timestep prediction proposed in our framework.

**Multi-agent regressive models.** Several prior works addressed modeling specific multi-agent problems via regressive models as one-off case studies. We introduce the PAR framework to unify these efforts into a single cohesive framework. Many behavior prediction works focus on two agents

engaging in social interaction, whether it be dyadic communication (Ng et al., 2022; 2023; 2024) or social dance (Siyao et al., 2024; Maluleke et al., 2024). These studies primarily tackle the challenge of predicting the state of an interacting partner (Person B) based on the input from Person A's state, sometimes extending predictions into the future (Guo et al., 2022; Maluleke et al., 2024). While earlier works used architectures such as variational RNNs (Baruah & Banerjee, 2020), recent works have predominantly adopted transformer architectures for social interaction modeling (Guo et al., 2022; Ng et al., 2022; Chopin et al., 2023; Ng et al., 2023; Siyao et al., 2024), with some works exploring diffusion (Liang et al., 2024), or diffusion with attention (Ghosh et al., 2024). Our PAR framework focuses on transformer models.

Works encompassed by the PAR framework extend beyond human social interaction. Many multi-agent human or car trajectory prediction approaches use autoregressive prediction. For instance, MotionLM (Seff et al., 2023) utilizes a transformer decoder that processes multi-agent tokens, incorporating a learned agent ID embedding. This methodology informs our approach across all our case studies. *Critically, in contrast to all prior multi-agent regressive works that all addressed specific applications, we demonstrate, for the first time, that we can unify a diverse set of seemingly different multi-agent regressive problems under a single PAR framework.*

**Action recognition/forecasting.** Recent advancements in action recognition have significantly improved our ability to understand and classify human activities in videos, starting with the SlowFast network (Feichtenhofer et al., 2019), which introduced a two-pathway approach that processes visual information at different frame rates to capture slow and fast motion patterns. This resembles ventral and dorsal pathways of human brain for action understanding and object recognition, respectively. With the introduction of transformers (Vaswani, 2017; Dosovitskiy, 2020), MViT (Fan et al., 2021) showed promising results on action understanding benchmarks with multi scale transformers. Recently, Hiera (Ryali et al., 2023), presented a hierarchical vision transformer that leverages multi-scale feature learning to enhance action recognition performance, by utlizing masked image pretraining as in MAE He et al. (2022). LART (Rajasegaran et al., 2023), expanded on these by incorporating 3D human pose trajectories and achieve better action prediction performance. (Sun et al., 2019) perform action forecasting on videos using relational information. (Loh et al., 2022) learn a RNN on long form videos, to contextualize the long past and make better predictions of the future.

**Car trajectory prediction.** Forecasting the future motion of cars is a popular problem in the space of autonomous vehicles (Huang et al., 2022; Cui et al., 2024), facilitated by an influx of datasets in recent years (Chang et al., 2019; Caesar et al., 2020; Sun et al., 2020). Many important approaches have focused on modeling the environment in conjunction with multiple agents (Casas et al., 2018; Cui et al., 2019; Salzmann et al., 2020); our framework only focuses on multi-agent interactions. More recent advancements have seen the rise of transformer-based methods in trajectory prediction (Ngiam et al., 2021; Yuan et al., 2021). In particular, MotionLM (Seff et al., 2023) forecasts multiagent trajectories by encoding motion in discrete acceleration tokens and passing these tokens through a transformer decoder that cross-attends to the Wayformer (Nayakanti et al., 2023) scene encoder. We use acceleration tokens to discretize car motion.

**6D pose estimation and hand-object interaction.** 6D pose estimation from monocular camera images has been extensively studied (Xiang et al., 2017; Li et al., 2018; Trabelsi et al., 2021; Wang et al., 2021). Additionally, a related area of research known as 6D object pose tracking leverages temporal cues to improve the accuracy of 6D pose estimation in video sequences (Wen et al., 2020; Deng et al., 2021; Wen et al., 2023; 2024). There is also significant interest in learning state and action information of hands and objects through hand-object interaction data, sourced from both curated and in-the-wild video data (Wu et al., 2024). Of particular relevance to 6D pose estimation is the DexYCB dataset (Chao et al., 2021), which contains 1000 videos of human subjects interacting with 20 objects on a table with randomized tabletop arrangements and 6D object poses. For the third case study in this paper, we propose using the poly-autoregressive framework to model hand-object interactions, demonstrating that incorporating the hand as an agent provides a useful prior for enhancing object rotation and translation predictions.

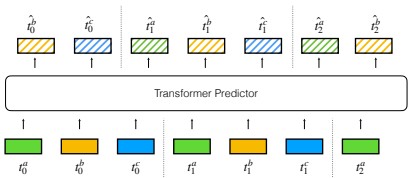 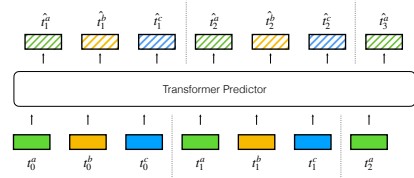

(a) AR: multi-agent, next-token training.     (b) PAR: same-agent, next-timestep training.

Figure 2: Training with teacher forcing for (a) multi-agent next-token prediction in autoregressive models and (b) multi-agent poly-autoregressive models. Solid indicates a ground-truth token and striped predicted. Color denotes agent identity. The AR model is trained for next-token prediction, while the PAR model is trained to predict the next timestep of the same agent.

## 3 POLY-AUTOREGRESSIVE MODELING

Our goal is to model the behavior of an agent while considering any other agents with whom they interact (if any). To test whether our model captures the dynamics of interaction, we predict the agent's future behavior and compare it to ground truth in a data-driven way.

We define the following task: *In an interaction setting of N agents, given the observed past states of N-1 agents, and the observed or previously-predicted past states of the Nth agent, predict the future states of the Nth agent.*

To represent the ongoing flow of interaction, we define a transformer-based poly-autoregressive (PAR) predictor, $\mathcal{P}$, that learns to model temporally long-range correlations in the input sequence. The inputs to the predictor are the past states of the N interacting agents, and its output is the predicted future state of the Nth agent.

### 3.1 PROBLEM DEFINITION

Let $\mathbf{S} = \{\mathbf{s}_i\}_{i=1}^T$ be a temporal sequence of agent states, $\mathbf{s}_i$. We use $\mathbf{S}^N$ and $\mathbf{S}^{1:N-1}$ to denote the temporal sequences of states of the $N_{th}$ agent and of the other $N-1$ agents, respectively. For each timestep $t \in [t_\pi, T]$, where $t_\pi \in [1, T]$ is the time in which we start predicting, we take as input all other $N-1$ agents' past observed state sequences $\mathbf{S}_{1:t-1}^{1:N-1}$ along with the $N_{th}$ agent's past observed states up to $t_\pi$, $\mathbf{S}_{1:t_\pi}^N$, and any of its previously predicted past states $\hat{\mathbf{S}}_{t_\pi+1:t-1}^N$, if available (see Figure 1b). Our predictor, $\mathcal{P}$, then *poly-autoregressively* predicts the $N_{th}$ agent's future states one time-step at a time:

$$\hat{\mathbf{s}}_t^N = \mathcal{P}(\mathbf{S}_{1:t-1}^{1:N-1}, \mathbf{S}_{1:t_\pi}^N, \hat{\mathbf{S}}_{t_\pi+1:t-1}^N), \tag{1}$$

where $\mathcal{P}$ learns to model the distribution over the next timestep of the $N_{th}$ agent's states, given the states of all other agents:

$$p(\hat{\mathbf{s}}_t^N | \mathbf{S}_{1:t-1}^{1:N-1}, \mathbf{S}_{1:t-1}^N). \tag{2}$$

While we provide the observed ground truth states of other agents at inference, during training, we jointly maximize the likelihood of all $N$ agents by computing losses on their future state predictions.

We train the predictor by maximizing the likelihood of the target state $y$ at time $t$:

$$\mathcal{L}_\mathcal{P} = \mathbb{E}_{y \sim p(y)}[-\log(p(\mathbf{s}_t^N))],$$

where the target state $y$ at $t$ is computed from the $N_{th}$ agent ground truth future state.

### 3.2 THE POLY-AUTOREGRESSIVE FRAMEWORK

We address the problem of forecasting the future states of an agent (from time $t$ to $T$) in a data-driven way, given a temporal sequence of past states (from time $1$ to $t-1$). We assume that our agent has some feature, or a set of features, of interest in a video (e.g., 3D pose) that we can tokenize. We predict the future states of the agent in terms of this tokenized feature (or set of), where we use one token (or set of tokens) per time step. The predicted tokens can be discrete (i.e., an index into a feature codebook) or continuous (i.e., a vector of one or more continuous values). The loss $\ell$ will

depend on the problem's specifics and the type of token used. To train the model to predict the future, we rely on all the interaction dynamics of length $T$ in our training dataset as ground truth examples.

As a baseline, we consider the **single-agent autoregressive (AR)** paradigm, where a transformer is trained to perform next-token-prediction with teacher forcing. AR uses greedy sampling to generate sequences at inference time, predicting one next token at a time (Figure 1a).

In contrast, our **multi-agent poly-autoregressive (PAR)** framework considers the other $N - 1$ agents in the scene when predicting the future state of the $Nth$ agent. In this setup, we tokenize the features of interest of all $N$ agents, yielding $N$ tokens at each timestep for a total of $N * T$ tokens. In practice, we operate on a flattened sequence of $N * T$ tokens. Rather than repeating the single-agent AR training procedure of next-token prediction in this multi-agent case (as in Figure 2a), we jointly model the $N$ agents at each timestep by introducing the following features to our PAR framework.

**Next-timestep prediction.** A standard autoregressive model predicts the next token. Given the flattened sequence of $N * T$ tokens our model operates on, next token prediction would take as input an agent $k$ at timestep $t$ and predict agent $k + 1$'s state at the same timestep $t$ (as in Figure 2a). However, our goal is to predict the input agent $k$'s future state at time $t + 1$. Therefore, we perform *same-agent next-timestep* prediction rather than next-token prediction (See Figure 2b for an illustration of same-agent next-timestep at training).

**Learned agent identity embedding.** When giving a model information corresponding to multiple agents, the model can benefit from knowing which token corresponds to which agent. We give the model this information with a learned agent ID embedding.

**Joint training.** We train the model to jointly predict the future of all agents by computing a loss on the predicted tokens of all agents (Figure 2b). Please refer to Section 3.1 for our inference paradigm.

### 3.3 TASK-SPECIFIC CONSIDERATIONS

While the PAR framework is simple, it unifies diverse problems under a single framework and architecture without any modifications. In order to formulate a problem as interaction-conditioned prediction in terms of the PAR framework, users must consider several task-specific details.

**Data.** The dataset naturally varies with the nature of the task. The input data source in our example tasks is always a collection of videos. From these videos, we extract various modalities relevant to the task at hand. These modalities can range from high-level features, such as action class labels, to low-level ones, such as 3D pose. We assume that each agent in the dataset is detected at each frame and is associated with an agent ID.

**Tokenization.** Our framework supports both discrete, quantized tokens and continuous vector tokens. The choice between discrete and continuous depends on the nature of the task. In the case of discrete tokens, we use a standard embedding layer to project to the hidden dimension. For continuous tokens, we train a projection layer to project the token into the hidden dimension of the transformer. For instance, if our continuous token is a 3D vector with an $(x, y, z)$ 3D location coordinate and our hidden dimension is 128, our projection layer will project from 3 to 128 dimensions. We also train an un-projection layer that reverts the hidden dimension to the original token dimension.

**Loss.** The type of token and task-specific considerations dictate the loss function $\ell$ applied during model training. For discrete tokens, a classification loss is appropriate. For continuous tokens, we use a regression loss on the original token dimension.

**Baselines.** We compare to the following baselines, where applicable on a case-by-case basis:

• *Random token*: pick random tokens from the best available token space and use as the prediction.

• *Random trajectory*: pick at random a trajectory from the training dataset to use as the prediction.

• *NN*: Given an input agent $A$'s trajectory history, find the closest trajectory to it in the training set, belonging to $A^T$. Use $A^T$'s future as the predicted future.

• *Multiagent NN*: In a dataset with two interacting partners $A$ and $B$, given an input agent $A$'s trajectory history, find the closest trajectory to it in the training set, belonging to $A^T$. Use $A^T$'s interaction partner's $B^T$'s future as the prediction.

• *Mirror*: In a dataset with two interacting partners $A$ and $B$, use the ground truth future of agent $B$ as the predicted future for agent $A$.

### 3.4 Framework Implementation Details

We keep the following implementation details constant for all case studies (see also Sec. A.1).

**Learned agent ID embedding.** Our learned agent ID embedding maps the integer ID of an agent to a hidden dim-sized vector. It is then summed to the token embedding and input to our model.

**Architecture.** For all case studies, we use the Llama (Touvron et al., 2023) transformer decoder architecture with 8 layers, 8 attention heads, and a hidden andintermediate dimension of 128. The decoder has ∼4.4M learned parameters, not including learned embedding layers which add a few thousand more parameters. A rotary positional encoding (Su et al., 2024) is used in addition to other summed encodings (i.e. agent ID embedding, locational positional encoding in Sec. 5). We train using teacher forcing. The only hyperparameter that changes between case studies is the learning rate.

## 4 Case Study 1: Social Action Prediction

Our first case study involves forecasting human actions. Human behaviors are fundamentally social; for instance, individuals frequently walk in groups and alternate between speaking and listening roles when conversing. Certain actions, like hugging or handshaking, are intrinsically multi-person. Therefore, modeling human interactions should help improve action prediction performance, especially on multi-person actions, which we show in this case study.

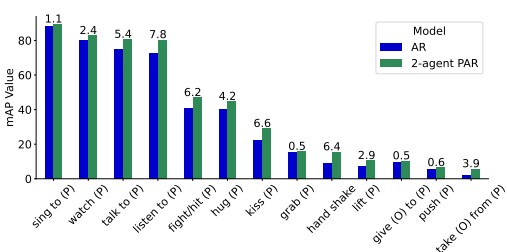

Figure 3: **Per-class mAP for AVA 2-person actions**. We see performance improvement on every 2-person AVA action class ((P) stands for "a person"). Some absolute mAP gains are particularly significant: listen to +7.8, kiss +6.6, hand shake +6.4, fight/hit +6.2, talk to +5.4, take from +3.9.

### 4.1 Experimental Setup

**Dataset.** The Atomic Visual Actions (AVA) dataset (Gu et al., 2018) comprises 235 training and 64 15-minute validation videos from movies. Annotations are provided at a 1Hz frequency, detailing bounding boxes and tracks for individuals within the frame, and each person's actions within a 1-second timeframe. Individuals may engage in multiple concurrent actions from a repertoire of 80 distinct action classes (e.g., sitting and talking simultaneously). For our analysis, we select clips featuring a continuous sequence of an agent's actions spanning at least 4s, splitting sequences exceeding 12s. We use the first half of each clip as history to predict the second half.

**Task-specific considerations.** Each agent's token $\mathcal{A}$ represents an 80-dimensional vector that corresponds to the actions performed at a specific timestep. Each element denotes the probability of a particular action class being enacted; ground-truth inputs are a binary vector. We implement an embedding layer that projects these tokens into the transformer's hidden dimension, as well as an un-projection layer that reverts them back to the original 80-dimensional token space for the purposes of loss calculation and output generation. We do not explicitly require the outputs to be values between 0 and 1. We use a MSE regression loss on the 80D action tokens: $\mathcal{L} = \frac{1}{n} \sum_{i=1}^{n} (\mathcal{A}_i - \hat{\mathcal{A}}_i)^2$. Our evaluation metric is the mean average precision (mAP) on the 80 AVA classes.

We implement all baselines described in 3.3, where *Random Token* corresponds to a random 80D vector sampled from 0 to 1. *NN* and *Multiagent NN* use Hamming distance as the distance metric.

### 4.2 Results

We report the performance of a single-agent AR model as a baseline, in the first line of Table 1a. The AR model is significantly better than our baselines (see Table 1b), the strongest baseline

| Agents | Timestep pred | Ag ID embd | mAP ↑ |
|---|---|---|---|
| 1 | N/A | N/A | 34.8 |
| 2 | ✗ | ✗ | 29.8 |
| 2 | ✗ | ✓ | 32.2 |
| 2 | ✓ | ✗ | 33.7 |
| 2 | ✓ | ✓ | **36.6** |

(a) PAR action prediction performance on AVA

| Baseline | Agents | mAP ↑ |
|---|---|---|
| Random Token | 1 | 3.46 |
| Random Training Traj | 1 | 3.44 |
| Nearest Neighbor | 1 | 13.17 |
| Multiagent NN | 2 | 5.10 |
| Mirror | 2 | 7.97 |

(b) AVA baselines

Table 1: **Action prediction on AVA a)** Without next-timestep prediction and learned agent ID embedding, our model struggles with multi-agent reasoning, performing worse than the AR baseline. With these PAR components, the 2-agent PAR model achieves a +1.8 mAP gain over the AR method (see Fig 7 and Fig 3 for class breakdown). **b)** While the nearest neighbor baseline performs best among baselines, it is still significantly worse than the AR model.

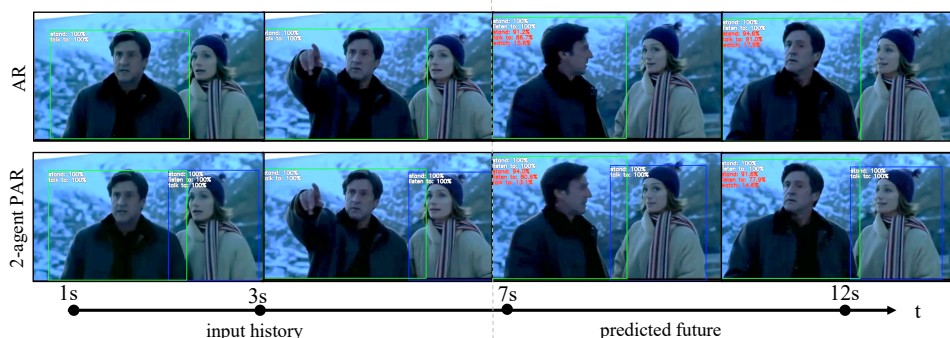

Figure 4: **Action prediction example.** The distribution over ground truth actions are in white, and our predictions in red. A 6s action history (1Hz) is input, and 6s of future actions are predicted. In the scene, the man and woman alternate between talking and listening. Initially, the man is talking. The AR model predicts the man will continue talking, while the 2-agent PAR model recognizes the woman is talking and predicts more accurate turn-taking behavior.

being the single-agent NN. We compare these baselines to our 2-agent PAR model (last line) and various ablations where we remove the agent ID embedding and perform next-token rather than same-agent next-timestep prediction. The second line of the table corresponds to multi-agent next-token prediction(Figure 2a). We see that this approach confuses the model, and the performance is significantly worse than just training on and considering a single agent. However, as we add various components of our PAR approach, the performance improves, and with both the next timestep prediction and agent ID embedding, we get a $1.8\%$ mAP gain.

In Fig. 4 we see an example of action prediction. In the input history, the man talks and the woman listens. In the future, the woman talks, and then man listens. Our 2-agent PAR model on the bottom row has that talking and listening actions are complementary actions, while the AR model does not make predictions that demonstrate this understanding. We see quantitative evidence of this in Fig. 7, with per-class mAPs for our AR vs 2-agent PAR model for 2-person action classes. Here, the category of talk to gets a +5.4 mAP gain and the category of listen to gets a +7.8 mAP gain when we train a multi-agent model. We also see a significant boost on many other interaction-related action classes - kiss a person +6.6, fight/hit a person +6.2, lift a person + 2.9, and take from a person +3.9.

## 5 CASE STUDY 2: MULTIAGENT CAR TRAJECTORY PREDICTION

Our second case study focuses on predicting car trajectories. Trajectory prediction requires a vehicle to be aware of other cars on the road to avoid collisions and promote cooperative behavior. This study demonstrates how our framework enables the joint modeling of multiple vehicles' movements.

| Token type | LPE | Agents | ADE ↓ | FDE ↓ |
|---|---|---|---|---|
| Velocity | ✗ | 1 | 1.50 | 3.64 |
| Velocity | ✗ | 3 | 1.45 | 3.51 |
| Accl | ✗ | 1 | 1.44 | 3.57 |
| Accl | ✗ | 3 | 1.40 | 3.44 |
| Accl | ✓ | 3 | **1.35** | **3.34** |

(a) PAR car trajectory prediction performance

| Baseline | Agents | ADE ↓ | FDE ↓ |
|---|---|---|---|
| Random Trajectory | 1 | 8.89 | 16.51 |
| NN | 1 | 1.80 | 4.13 |
| Multiagent NN | N | 6.40 | 12.04 |
| Mirror | N | 11.59 | 14.93 |

(b) Car trajectory prediction baselines

Table 2: **Car trajectory prediction on nuScenes a)** Comparing 3-agent PAR and single-agent AR with velocity and acceleration tokens shows stronger performance with acceleration tokens for both models. Adding location via positional encoding (LPE) further improves results. **b)** Nearest neighbor performs best overall, but our learned single-agent AR models outperform all baselines.

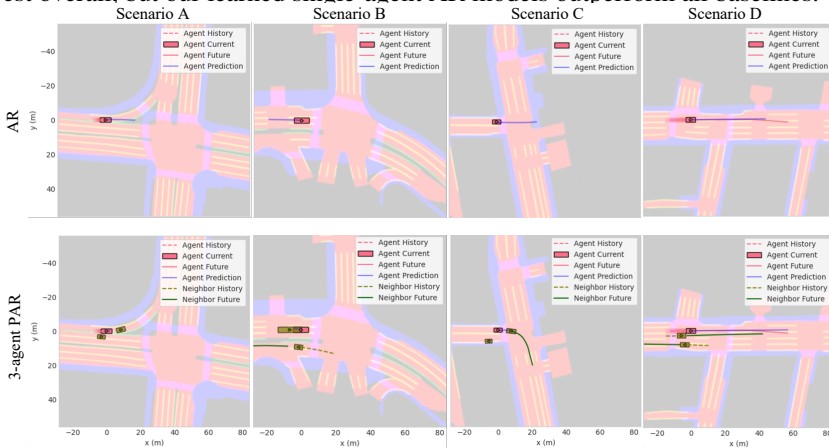

Figure 5: Example results from our single-agent AR model (top row) and three-agent PAR model with location positional encoding (bottom row) on nuScenes. The predicted agent's ground truth trajectory is in pink, and the predicted future in blue. For the PAR model, the other two agents' ground truth states are in green. Qualitatively, the PAR model handles situations where single-agent predictions might lead to collisions (A, B), uses other agents' behavior to better adhere to road areas (A, C) without environment data, and predicts based on the speed changes of other cars (D).

## 5.1 EXPERIMENTAL SETUP

**Dataset.** We use nuScenes (Caesar et al., 2020) , inputting 2 seconds of position data to forecast vehicle positions 6 seconds ahead. Specifically, our objective is to predict the future $xy$ coordinates of each vehicle. Our analysis exclusively considers vehicles as agents. We use the `trajdata` interface (Ivanovic et al., 2023) to load and visualize the dataset.

**Task-specific considerations.** Instead of discretizing the $xy$ position space, we discretize the motion, resulting in discrete velocity or acceleration tokens. These integer tokens are projected to the transformer hidden dimension using the Llama token embedding layer. Inputting only these tokens results in our PAR model knowing what speed the other agents are going at, but not where they are. It is important the model has this awareness (it should know if two agents are going to collide), so our model needs to reason over this second modality of location. We implement this by passing locations relative to the agent we are predicting into a sin-cosine positional embedding (see details in Sec. A.2), which we denote a location positional encoding (LPE). The LPE is summed to our token embeddings.

We use a cross-entropy classification loss on our discrete tokens: $\mathcal{L} = \mathbb{E}_{y \sim p(y)}[-\log(p(\mathbf{s}_{t+1}^N))]$. We use the standard average displacement error (ADE) and final displacement error (FDE) to evaluate our predicted trajectories. For our baselines (Sec. 3.3), we use the closest agent at the current timestep for *Multiagent NN* and *Mirror*. For *NN* and *Multiagent NN* we use MSE as the distance metric.

## 5.2 RESULTS

We train AR and 3-agent PAR models using velocity tokens, acceleration tokens, and acceleration tokens combined with our location positional encoding. The results can be seen in Table 2a. Note

that the 3-agent PAR model uses the agent ID embedding and next timestep prediction. Acceleration tokens consistently outperform velocity tokens both for agent AR and 3-agent PAR models. This could be because the vocabulary size for acceleration tokens is much smaller and therefore easier to optimize. Regardless, both ways of tokenizing result in models that outperform our baselines (see Table 2b - NN has a relatively low error on this dataset), and highlight that our framework is flexible such that a user can experiment with different ways of representing entities. For both token types, the 3-agent PAR model that is blind to location outperforms the AR model. While location information should help the model, it is possible that simply knowing whether other agents are slowing down or accelerating can help the model make better predictions. When adding location information via the LPE to our 3-agent PAR model, we see another performance gain in ADE and FDE.

Qualitative examples of the AR model (top row) and 3-agent location-aware PAR model (bottom row) can be seen in Figure 5. Our method uses no image or environment data (e.g., lanes) as input. However, by reasoning over multiple agents, its predictions lead to fewer collisions and better reasoning about speed changes and driveable areas based solely on other agents' behaviors.

# 6 CASE STUDY 3: OBJECT POSE ESTIMATION DURING HAND-OBJECT INTERACTION

Our final case study explores how hand-object interaction can be leveraged for object pose estimation. We conceptualize the human hand and the interacting object as two agents, with tokens representing distinct state types. We show that our PAR framework allows us to jointly model these agents, improving our ability to predict the object's 3D translation and rotation.

## 6.1 EXPERIMENTAL SETUP

**Dataset.** For this case study, we utilize the DexYCB dataset Chao et al. (2021), which contains 1000 videos of 10 human subjects performing object manipulation tasks. Each subject picks up 20 distinct objects from the YCB-Video dataset Xiang et al. (2017), with multiple trials conducted for each object. The dataset is divided into 800 training videos, 40 validation videos, and 160 testing videos. Although the videos are recorded from 8 RGB-D cameras, we work with a single camera view. In each trial, the subject starts in a relaxed pose with their hand by their side (often out of the camera's view), grasps the target object, and lifts it into the air. For each subject-object pair, there are 5 trials where the object's rotation, placement, and surrounding distractor objects are randomized. The dataset provides labels such as the object's SO(3) rotation and 3D translation, and the 3D positions of 21 hand joints in camera space. We focus on predicting either the object's rotation or translation as it is being picked up in each video.

**Task-specific considerations.** In object-only experiments, we tokenize object information, while in hand-object experiments, both object and hand information are tokenized. The object is represented as 4-dimensional tokens for rotation-only prediction (quaternion for SO(3) rotation) or 3-dimensional tokens for translation-only prediction (Euclidean coordinates). In hand-object experiments, the hand is represented by a 63-dimensional vector corresponding to 21 hand joints, and agent ID embeddings distinguish between the hand and object. An embedding layer projects the tokens into the transformer's hidden dimension, and another layer projects them back for loss computation and generation. Teacher forcing is applied during training, with hand joint information teacher-forced in validation while generating the object's rotation or translation. For rotation-only prediction, the loss is $\mathcal{L}_{rot} = 1 - |\hat{q} \cdot q|$, where $\hat{q}$ is the predicted quaternion and $q$ the ground-truth quaternion. For translation-only prediction, the loss $\mathcal{L}_t$ is the mean squared error (MSE) between predicted and ground-truth translations. In hand-object experiments, the additional loss $\mathcal{L}_h$ is the MSE on hand joint positions. The object-only rotation model is optimized with $\mathcal{L}_{rot}$, while the hand-object rotation model combines $\alpha\mathcal{L}_{rot} + (1 - \alpha)\mathcal{L}_h$ where $\alpha = 0.33$; similarly, the object-only translation model is trained with $\mathcal{L}_t$, and the hand-object translation model uses $\mathcal{L}_t + \mathcal{L}_h$. For validation, the first half of each video is provided, and object predictions are autoregressively generated for the second half. Translation performance is evaluated using MSE, while rotation is measured using geodesic distance (GEO) on SO(3), computed by converting quaternions to SO(3) matrices.

| Type | Object Token | Hand Token | Ag ID Emb | Agents | MSE ($m^2$) ↓ | GEO ($rad$) ↓ |
|------|:---:|:---:|:---:|:---:|:---:|:---:|
| Translation | ✓ | ✗ | ✗ | 1 | $1.2 \times 10^{-2}$ | - |
| Translation | ✓ | ✓ | ✓ | 2 | $\mathbf{8.6 \times 10^{-3}}$ | - |
| Rotation | ✓ | ✗ | ✗ | 1 | - | 1.03 |
| Rotation | ✓ | ✓ | ✓ | 2 | - | **0.88** |

Table 3: **Test set results on DexYCB dataset.** Top two rows: translation prediction, bottom two rows: rotation prediction. In both cases, the 2-agent PAR model, which accounts hand-objectinteraction by integrating the hand as an additional agent, yields improved results.

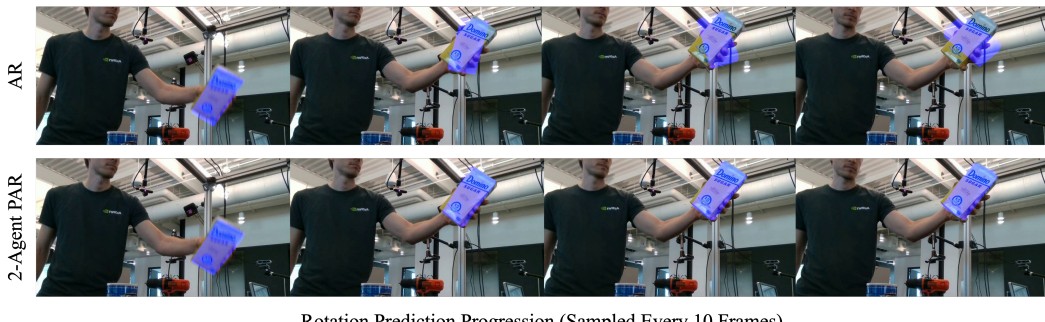

Rotation Prediction Progression (Sampled Every 10 Frames)

Figure 6: **Rotation prediction qualitative result.** The projected 3D model in blue has the ground-truth translation for visualization purposes and our predicted rotation. To account for the low dynamics between consecutive frames, we sample every 10th frame. In the top row (AR), the results depict the object of interest as the sole agent, while the bottom row (2-agent PAR) demonstrates improved performance by incorporating the human hand as a second agent in the grasping interaction.

## 6.2 RESULTS

For both rotation-only and translation-only predictions, the object-only models serve as baselines for comparison with the hand-object PAR models. Refer to Table 3 for the quantitative results of the two prediction tasks, and Figure 8 for quantitative results on the rotation prediction task. In both prediction tasks, we observe that incorporating the human hand's interaction with the object enhances accuracy. In Figure 8, we see that the AR model (top row) achieves high-fidelity predictions early on, when much of its history still relies on ground truth data from the first half of the sequence. However, as the video progresses and the history becomes increasingly dependent on predicted object rotations, the AR model's performance rapidly deteriorates. In contrast, our PAR model (bottom row) reasons over the 3D hand joint positions to predict the object's SO(3) rotation much more accurately.

## 7 DISCUSSION

This work introduced the Poly-Autoregressive (PAR) modeling framework, a unifying approach to prediction on interacting entities. By applying the same transformer architecture (and hyperparameters) across diverse tasks such as action prediction in social settings, trajectory prediction for autonomous vehicles, and object pose prediction during hand-object interaction, we have demonstrated the versatility of our framework.

Our findings underscore the crucial importance of considering the influence of multiple agents in a scene. By modeling interactions, we significantly improved prediction accuracy compared to traditional single-agent approaches on all three problems we considered. While we achieved promising results with a simple architecture, there is ample room for improvement in future work. Incorporating environmental context is another important avenue for future research.

The simplicity and generalizability of our PAR framework presents a strong foundation, offering universal building blocks that can be extended or refined for future tasks. The potential for future advancements in AI systems that can more accurately navigate and operate within real-world environments fall under the PAR framework is significant, marking an important step in moving towards prediction in the real world.

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

# A APPENDIX

## A.1 ADDITIONAL PAR FRAMEWORK IMPLEMENTATION DETAILS

**Token embeddings and loss.** For discrete tokens, we use a standard learned embedding layer to convert the tokens to the hidden dimension $d_h$ of the model. To compute the loss, we use a classification loss between the predicted distribution (output logits) and the input ground truth tokens. For continuous inputs with dimension $d$, we learn a linear layer to project from $d$ to $d_h$, and a second un-projection layer to project from $d_h$ back to $d$. To compute the loss, we take the last hidden state of the model, un-project it back to $d$, and then compute a regression loss in the original token space.

**Next-timestep prediction.** In standard autoregressive models (such as our single-agent model in section 3.2) the next token prediction objective is enforced by computing the loss on an input and predicted target that are both shifted by one. Now, we will instead shift both by $N$, so that for a given token, the model operating on our flattened sequence of $N * T$ tokens predicts a token corresponding to the next timestep but the same agent.

**Inference.** For a single-agent model, starting with an initial sequence history of $h$ tokens, we feed these into the model to get the next token, which we then append to our sequence to form a new sequence of $h + 1$ tokens. We repeat this process to generate arbitrarily long sequences.

For our multi-agent model, we start with a ground-truth history of $h$ timesteps, which corresponds to $h * N$ tokens, including the ego agent, agent $N$. Inputting this to the model results in the last output

token being our ego agent at timestep $h + 1$. Then, to predict the next timestep $h + 2$, we concatenate to the ground truth $h * N$ tokens the ground truth of agents $1 : N - 1$ at timestep $h + 1$ and our prediction of the ego agent at timestep $h + 1$, and we repeat this process.

For a multiagent next-token prediction ablation, to predict the ego agent at timestep $h + 1$, we feed in the ground truth of agents $1 : N - 1$ at $h + 1$ to our model to predict our ego agent, agent $N$, at timestep $h + 1$. We continue this process of giving our model the ground truth tokens of agents $1 : N - 1$ to predict agent $N$ at each timestep.

## A.2 ADDITIONAL CAR IMPLEMENTATION DETAILS

**Tokenization** Instead of discretizing the $xy$ position space, we discretize the motion, resulting in discrete velocity or acceleration tokens computed as follows. We take each agents ground truth trajectory (past and future), shift it so that the trajectory starts at $x, y = 0, 0$, and then rotate the trajectory such that its initial heading at $t = 0$ is 0 radians. We divide velocity space into 128 even bins in $[-18, 18]$ meters. We then, separately for $x$ and $y$, take the difference between each pair of coordinates in the trajectory, to get a length $T - 1$ sequence of deltas. Each of these deltas is mapped to a bin index.

We first experimented with velocity tokens, taking the Cartesian product of bin space to give each $xy$-delta one single integer index between 1 and $128 * 128 = 16384$. To get acceleration tokens, we take the difference between each $x$ delta and $y$ delta, and bin these differences into 13 bins. We then take the Cartesian product of bin space to get a vocabulary between 1 and $13 * 13 = 169$.

**Location Positional Encoding (LPE)** We implement our location positional encoding as follows.

We first compute relative location to the agent we are predcting (the "ego" agent) at the first timestep of the history. The ego agent trajectory is shifted to be at location $(0, 0)$ at time $t = 0$, and all other agents are shifted to be relative to the ego agents position. We also rotate the ego agent trajectory to have a heading of 0, and rotate all other agents trajectories relative to this ego agent trajectory.

We normalize these relative locations (in meters) to be between 0 and 1. We then quantize these normalized locations to be an integer between 0 and 100. We next pass these locations (x and y separately) into a sin-cos positional encoding. Instead of operating on sequence position indices, the positional encoding operates on the quantized locations. We compute separate positional encodings for x and y. We either have these encoding dimensions be half of the hidden dimension so we can concatenate, or we sum the x and y encodings to get one encoding. We then sum the result of this encoding to the model inputs at training for the full trajectory (history and future).

At inference, we compute this encoding on the full trajectory (history and future) for agents 1 to N-1, but for our ego agent, we only use the history location ground truth. To get the future locations, at each sampling step, we integrate over our velocity or acceleration token to update the predicted location one step at a time, and then pass that location into our encoding.

**Evaluation dataset** Since the nuScenes test set can only be evaluated by submitting to the leaderboard, but we are interested in demonstrating the effectiveness of PAR over AR, we evaluate on the nuScenes validation set.

## A.3 ADDITIONAL RESULTS ON ACTION FORECASTING CASE STUDY

We see the results of our AR and 2-agent PAR methods on the AVA 1-person classes in Fig. 7. On the vast majority of these classes, our 2-agent PAR method is still stronger than AR. This is likely because there are many actions that people carry out together, whether it be 2 people both dancing (+1.2), walking together (+10.8), watching TV (+4.4), or listening to music (+7.3).

The AVA test set annotations are not released. Since we are focused on action forecasting from ground-truth past annotations instead of predicting actions from video frames, we evaluate on the validation set.

## A.4 ADDITIONAL RESULTS ON OBJECT POSE ESTIMATION

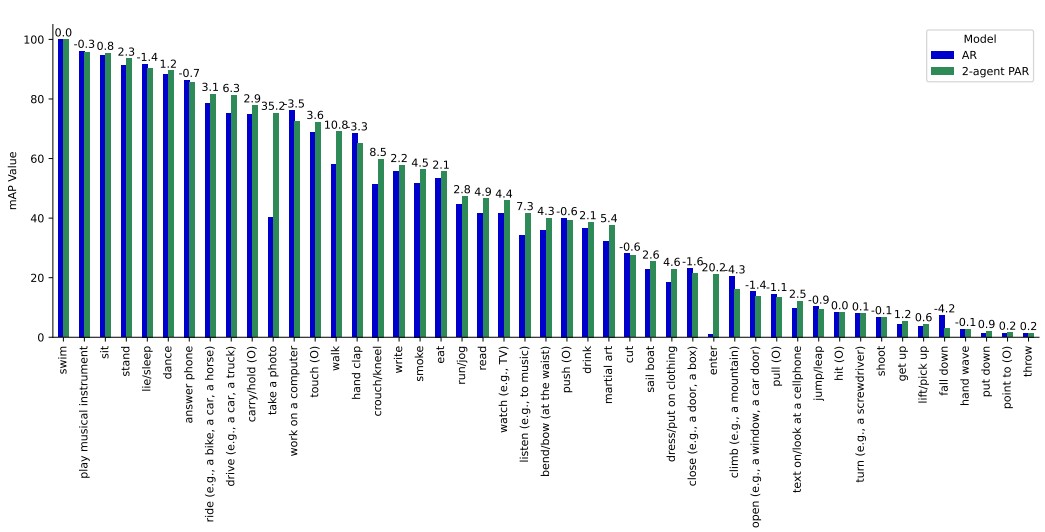

Figure 7: **Per-class mAP on AVA 1-person actions**. On these actions, our PAR method is still stronger for many classes. For instance, we get an absolute 10.8 mAP gain on walking - people often walk in groups, so it makes sense that this action would benefit from our PAR method.

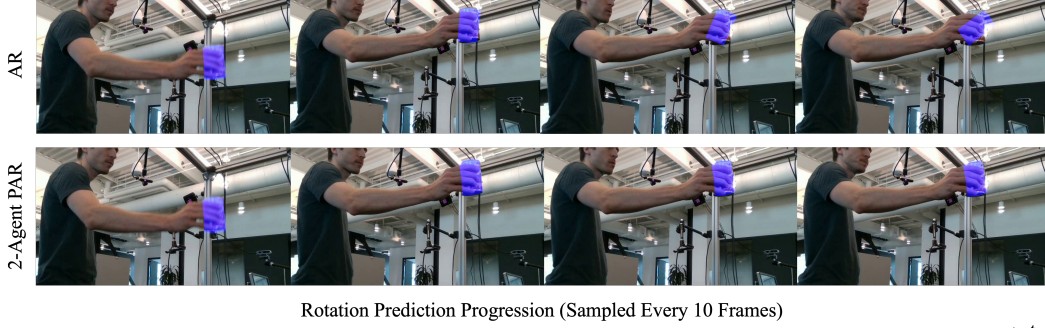

Rotation Prediction Progression (Sampled Every 10 Frames)

Figure 8: **Rotation prediction qualitative result.** The projected 3D model in blue has the ground-truth translation for visualization purposes and our predicted rotation. In the top row (AR), the results depict the object of interest as the sole agent, while the bottom row (2-agent PAR) demonstrates improved performance by incorporating the human hand as a second agent in the grasping interaction.

