# OpenReview forum: "Poly-Autoregressive Modeling for Interacting Entities"
_ICLR.cc/2025/Conference — ICLR 2025 Conference Withdrawn Submission_

### Official Review · Reviewer_gkm6 · 2024-11-04

**Soundness:** 2
**Presentation:** 2
**Contribution:** 1
**Rating:** 1
**Confidence:** 4

**Summary:**

The paper works on an already well established problem of predicting future states in scenarios involving multiple interacting agents. The model builds on traditional autoregressive approaches by incorporating interactions between agents. The authors frame behavior prediction as a multi-agent sequence generation task and implement their approach using a transformer-based architecture. The paper uses three distinct case studies: human action prediction, car trajectory forecasting, and object pose estimation during hand-object interaction. Limited Results presented in the paper  indicate that the PAR model achieves good performance compared to single-agent. Although no result against baselines is shown.

**Strengths:**

- The paper is well-structured and clearly written.
- The problem formulation and description are comprehensively detailed.

**Weaknesses:**

- Firstly the problem is not novel, as the authors have tried to portray in the paper. There are multiple graph and transformer based paper that handle multiagent prediction since 2017.
- The introduction fails to discuss relevant prior works comprehensively, making it difficult to discern the model's unique contributions.
The related works section is disorganized and lacks a critical analysis of how this work builds upon or diverges from existing methods.

- **No comparison against baselines**: The results section is inadequately supported by benchmarks; it lacks comparisons with established baselines. A quick search yields over 100 potential comparative studies on Transformer Multi-Agent Prediction, including graph-based methods known for their effectiveness in multi-agent contexts. https://scholar.google.com/scholar?hl=en&as_sdt=0,11&q=transformer+multi+agent+prediction

- All the three scenarios experimented in the paper could be handled by any of the existing paper like Agentformer or any graph based methods. I do not understand why authors claims that their work is novel

**Questions:**

See the weaknesses.

---

### Official Review · Reviewer_uGQn · 2024-11-09

**Soundness:** 2
**Presentation:** 2
**Contribution:** 2
**Rating:** 3
**Confidence:** 3

**Summary:**

This paper presents a unified prediction framework, PAR, which models agent interactions in dynamic scenarios by predicting an agent's future behavior based on temporal sequences of both the agent's and interacting partners' past states. PAR tokenizes agent features and predicts future states, making it adaptable to diverse predictive tasks such as social action prediction, car trajectory prediction, and object pose estimation.

**Strengths:**

The paper is well-written and includes a comprehensive evaluation across different tasks. The approach is straightforward and easy to understand.

**Weaknesses:**

1. The novelty of proposed framework is unclear. For each specific task, inputs are tokenized differently and then fed into PAR, which seems similar to other transformer-based methods that model interactions [1-5]. How does this approach differ from these methods? Is it simply a model that could be integrated into any prediction framework?

2. The paper lacks baseline comparisons for specific tasks. Each of the three tasks (social action prediction, trajectory prediction, and object pose estimation) is extensively studied in computer vision, with many established methods. Including comparisons with other relevant methods such as [1-5] for each task would strengthen the evaluation.

3. In trajectory prediction tasks, it’s unclear how interacting agents are identified. Identifying which agents influence others is a well-known issue in multi-agent trajectory prediction literature [5, 6]. How does the causal graph evolve with time, and how are interaction partners (e.g., Agent A’s interacting partners) identified? It would be helpful if the authors could clarify their approach to identifying interacting agents and explicitly mention the limitations of this approach.

4. Figure 1 does not provide a detailed overview of the model. Adding more detailed information on the model's overall functioning for at least one task would improve the paper's clarity.


References:

[1] Bae, Inhwan, Junoh Lee, and Hae-Gon Jeon. "Can Language Beat Numerical Regression? Language-Based Multimodal Trajectory Prediction." Proceedings of the IEEE/CVF Conference on Computer Vision and Pattern Recognition. 2024.

[2] Pazho, Armin Danesh, et al. "VT-Former: An Exploratory Study on Vehicle Trajectory Prediction for Highway Surveillance through Graph Isomorphism and Transformer." Proceedings of the IEEE/CVF Conference on Computer Vision and Pattern Recognition. 2024.

[3] Zhao, Qi, et al. "Antgpt: Can large language models help long-term action anticipation from videos?." arXiv preprint arXiv:2307.16368 (2023).

[4] Ausserlechner, Philipp, et al. "Zs6d: Zero-shot 6d object pose estimation using vision transformers." 2024 IEEE International Conference on Robotics and Automation (ICRA). IEEE, 2024.

[5] Sun, Qiao, et al. "M2i: From factored marginal trajectory prediction to interactive prediction." Proceedings of the IEEE/CVF Conference on Computer Vision and Pattern Recognition. 2022.

[6] Rowe, Luke, et al. "Fjmp: Factorized joint multi-agent motion prediction over learned directed acyclic interaction graphs." Proceedings of the IEEE/CVF Conference on Computer Vision and Pattern Recognition. 2023.

**Questions:**

1. What is the contribution of this paper, and how this is novel as compared to other methods in literature for each of the 3 tasks, for instance the comparison with methods such as [1-5] and explicitly stating the novelty and contributions as compared to these existing approaches. How does this approach differ from other approaches that model agents' interaction to predict future behavior?

2. Figure 1 doesn't provide a detailed overview of the model. Providing a detailed information on the overall working of the model with clear inputs, tokenization process, and predicted out for at least 1 task in the figure 1 would improve the clarity of the paper.

---

### Official Review · Reviewer_mxdr · 2024-11-13

**Soundness:** 2
**Presentation:** 2
**Contribution:** 1
**Rating:** 3
**Confidence:** 3

**Summary:**

The paper presents a method of training transformers with multi-agent state trajectories, which they Poly-Autoregressive (PAR) modelling. Tokens representing agent state can be discrete or continuous depending on the domain. The authors demonstrate PAR's ability to predict agent behaviour in three different domains: human action prediction, autonomous vehicle trajectory prediction and object pose prediction.

**Strengths:**

* The paper is well written and presented.

* PAR appears to be a simple but effective generic method to train multi-agent state trajectories.

* The authors demonstrate that PAR superior to the single-agent AR method and the multi-agent next-token AR method for predicting future agent states on the given datasets.

* The use of location position encoding (LPE) to encode relative positions between different agents appears to be novel and interesting method to encode such information for transformers.

**Weaknesses:**

* The effectiveness of PAR cannot really be assessed as it is only compared against weak baselines and a simple autoregressive single-agent mode.

* To truly assess the capabilities of PAR it needs to be compared to alternative SOTA methods for multi-agent state prediction.
  1. Comparison against generic multi-agent prediction methods such as variational RNNs, diffusion models and other transformer methods (as mentioned in the related work section)
  2. Comparison against specific methods for the three domains where the authors evaluated PAR, i.e. social action prediction, trajectory prediction, and object pose estimation.

* The claim that "in contrast to all prior multi-agent regressive works that all addressed specific
applications, we demonstrate, for the first time, that we can unify a diverse set of seemingly different
multi-agent regressive problems under a single PAR framework." is somewhat misleading. This is only true due to the domain-specific preprocessing and feature extraction (including domain-specific location-position encodings) that needs to be done in order to represent the agent state in a useful tokenised format.

**Questions:**

Other than the weaknesses that need addressing above.

Questions:
* Case study 2 does not appear to have been test with multi-agent AR, is this right?

Suggestions:
* The key information in figure 2 is so tiny I had to get a magnifying glass out just to tell that there was indeed a difference between diagrams (a) and (b)
* It is not clear in the tables which rows relate to single-agent AR, multi-agent AR and PAR.

---

### Note · Authors · 2024-11-13

I have read and agree with the venue's withdrawal policy on behalf of myself and my co-authors.